# Optimal Feature Analysis for Identification Based on Intracranial Brain Signals with Machine Learning Algorithms

**DOI:** 10.3390/bioengineering10070801

**Published:** 2023-07-04

**Authors:** Ming Li, Yu Qi, Gang Pan

**Affiliations:** 1State Key Lab of Brain-Machine Intelligence, Hangzhou 310018, China; 2College of Computer Science and Technology, Zhejiang University, Hangzhou 310027, China; 3Affiliated Mental Health Center & Hangzhou Seventh Peoples Hospital, MOE Frontier Science Center for Brain Science and Brain-Machine Integration, Zhejiang University School of Medicine, Hangzhou 310030, China

**Keywords:** biometrics, brain decoding, electroencephalogram, identification, intracranial brain signals, local field potential

## Abstract

Biometrics, e.g., fingerprints, the iris, and the face, have been widely used to authenticate individuals. However, most biometrics are not cancellable, i.e., once these traditional biometrics are cloned or stolen, they cannot be replaced easily. Unlike traditional biometrics, brain biometrics are extremely difficult to clone or forge due to the natural randomness across different individuals, which makes them an ideal option for identity authentication. Most existing brain biometrics are based on an electroencephalogram (EEG), which typically demonstrates unstable performance due to the low signal-to-noise ratio (SNR). Thus, in this paper, we propose the use of intracortical brain signals, which have higher resolution and SNR, to realize the construction of a high-performance brain biometric. Significantly, this is the first study to investigate the features of intracortical brain signals for identification. Specifically, several features based on local field potential are computed for identification, and their performance is compared with different machine learning algorithms. The results show that frequency domain features and time-frequency domain features are excellent for intra-day and inter-day identification. Furthermore, the energy features perform best among all features with 98% intra-day and 93% inter-day identification accuracy, which demonstrates the great potential of intracraial brain signals to be biometrics. This paper may serve as a guidance for future intracranial brain researches and the development of more reliable and high-performance brain biometrics.

## 1. Introduction

Identification can be classified into three groups: something the user knows (e.g., passwords), something the user has (e.g., ATM cards), something the user is (e.g., biometrics) [1]. Traditional methods in the first two categories, like passwords and ATM cards, have demonstrated obvious drawbacks. They can be forgotten, lost, or stolen, leading to unsuccessful authentication or information leakage [2]. Biometrics, which fall into the third category, can overcome these drawbacks. They do not require memorization because they are innate physiological or behavioral parts of the individual. They cannot be lost or stolen for the same reason. Due to the biological uniqueness of individuals, biometrics contain rich information to guarantee authentication security and biometrics are considered an ideal method to validate authorized users [3].

Conventional biometrics, such as fingerprint [4,5], face [6], iris [7], and DNA [8], have been extensively studied and widely adopted in real-life scenarios [9]. However, they each possess their weaknesses [10,11,12,13]. For instance, DNA can be easily stolen from any surface a target has touched; fingerprints can be faked through various methods, such as plastic molds and latex milk; faces can be forged by 2D pictures and high-resolution photography. Additionally, these traditional biometrics are not cancellable, meaning that if they are stolen, they cannot be replaced. A more secure biometric would meet two criteria: it would be more difficult to steal and it would be cancellable. Recent studies have demonstrated that the human brain can provide superior revocable biometrics [14,15,16,17,18,19]. In this case, brain electrical activity may meet above criteria, offering a more secure and potentially cancellable biometric alternative.

Most existing brain biometrics are based on electroencephalograms (EEG), which is collected above the scalp and is a type of non-invasive brain signals. Although EEG has shown high individuality among different people [20], which proves its potential as a biometric [21], existing EEG-based methods suffer from poor performance due to several issues:**(1) Low signal-to-noise ratio (SNR):** The electrical signals in the brain decay significantly as they pass through the skull and scalp, leading to low signal-to-noise ratio. This results in insufficient reliability and limited revocability for EEG-based systems.**(2) Unsatisfactory long-term stability:** Previous studies have identified a significantly decreasing trend in EEG performance over time [22].

To improve brain-based biometrics, determining how to achieve high reliability and long-term stability is an essential but challenging problem that needs to be addressed.

Intracortical brain signals, recorded with electrodes placed directly on the cortex reducing the signal attenuation, offer higher resolution and signal-to-noise ratio (SNR) compared to EEG [23,24], making them a promising option for constructing high-performance brain biometrics. Additionally, different from the EEG devices which are taken on before the experiment and taken off after the experiment, the electrodes for collecting intracranial brain signals are continuously implanted in the brain issues. For each take-on and take-off behavior of the EEG devices, the electrode impedance between the EEG electrode and the scalp could change significantly leading to the obvious signal noise, which is eliminated by the implanted electrodes of intracranial brain signals. To the best of our knowledge, this is the first study to investigate the features of intracortical brain signals for identification. In this paper, three groups of features are analyzed: time domain features, frequency domain features and time-frequency domain features. We also utilize 5 different classifiers to compare the performance of these features. The results show that frequency features and time-frequency features are excellent for intra-day and inter-day identification. In addition, energy features perform best among all features. This study can serve as a guidance for future intracranial brain researches and the development of more reliable and high-performance brain biometrics.

## 2. Methods

### 2.1. Brain Biometric Identification System

As shown in Figure 1, brain biometric systems typically consist of two parts: data acquisition part and decision-making part. The data acquisition stage involves capturing brain electrical activity using electrodes while the subject engages in certain paradigms. The collected data is then digitized and sent for preprocessing to enhance signal quality. Once the feature set has been extracted, biometric computations are performed using either simple statistical analyses or more complex machine learning approaches such as Neural Network (NN) or Support Vector Machine (SVM). The output of the system will be the identity label of the user for identification. Classifiers can combine the training module and identification module into one module, allowing them to complete both the matching score calculation and decision-making. It is important to note that the collected brain signals are usually contaminated with different kinds of noise and has a relatively low signal-to-noise ratio (SNR). Therefore, signal preprocessing is necessary to enhance signal quality before feature extraction and biometric computations.

### 2.2. Preprocessing

Local field potential (LFP) is a type of intracranial brain signals, which is collected by implanted microelectrodes in the midst of the population of neurons. Compared with EEG, LFP has the advantages of higher resolution and is more stable due to the fixed electrodes in the brain. The raw LFP is typically contaminated with electrical artifacts and the most common of these are 50 Hz noises from nearby electronics and muscular artifacts from the movements of the body. Therefore we preprocess the LFP to reduce or remove these artifacts in order to improve signal quality. Additionally, brain waves can be divided into five frequency bands:**Delta waves**: 0.5–4 Hz, associated with deep sleep and unconscious processes**Theta waves**: 4–8 Hz, related to drowsiness, light sleep, and some meditative states**Alpha waves**: 8–13 Hz, linked to relaxed wakefulness, eyes closed, and a calm state of mind**Beta waves**: 13–30 Hz, associated with active thinking, problem-solving, and focused attention**Gamma waves**: >30 Hz, related to high-level cognitive processing, memory, and perception

To capture more effective information based on above frequency bands, we employ a 0.5–300 Hz band-pass filter (i.e., a two-order Butterworth filter) and a 50-Hz notch filter to preprocess the raw LFP signals and we split the signals into 2-s long samples for feature extraction.

### 2.3. Feature Extraction

Feature extraction is a crucial stage in the processing and analysis of LFP signals, as the quality of the extracted features directly impacts the performance of the identification system. These features can be classified into three groups of domains: time domain, frequency domain, and time-frequency domain.

#### 2.3.1. Time Domain

The Autoregressive (AR) model is indeed a widely used time-domain feature in brain biometrics. The AR model represents a type of random process in which the output variable depends linearly on its own previous values and a stochastic term (an imperfectly predictable term). In the context of LFP signals, the AR model can be used to capture the temporal dependencies and patterns within the data. The general form of an AR model of order *p* is:(1)$$X_{t}=c+\sum_{i=1}^{p}a_{i}X_{t-i}+e_{t}$$
where $X_{t}$ is the output variable at time *t*; *c* is a constant term; $a_{i}$ are the coefficients of the model; $X_{t-i}$ are the previous values of the output variable; $e_{t}$ is the stochastic term (also known as the error term or residual) at time *t*. By fitting an AR model to LFP signals, you can extract coefficients as features that represent the underlying dynamics of the brain activity, which can then be used for biometric identification or other applications.

There are two common methods for the realization of AR models: Yule-Walker method and Burg’s method. Yule-Walker method applies a *p*-th order AR model to the windowed input signal by minimizing the forward prediction least square error and solving the autoregressive parameters directly. Differently, Burg’s method estimates the parameters using the Levinson-Durbin algorithm based on the last autoregressive-parameter estimated from each model order *p* by minimizing both the forward and backward prediction error. Compared to the Yule-Walker method, Burg’s method is often preferred for its lower computational complexity when estimating the parameters of an AR model [25]. Additionally, to determine the optimal order *p* of the AR model, there are generally three methods [26,27,28,29]:**Minimizing the error of the predictor equation through experimental results with different orders**: This method involves testing different orders of the AR model and selecting the one that results in the lowest prediction error.**Minimizing the Akaike Information Criterion (AIC)**: AIC is a measure of the goodness of fit of a statistical model that takes into account the number of parameters used:
(2)AIC(p)=Nlogε(p)+2p
where ε(p) is the modeling error and *N* is the length of the signal. The term 2p represents the penalty for selecting higher order models. By minimizing the AIC, we can find the optimal order of the AR model that balances model complexity and prediction accuracy.**Based on the eigenvalues of the matrix in the Yule-Walker equations**: The Yule-Walker equations are the following set of equations:
(3)$$\gamma_{m}=\sum_{k=1}^{p}a_{k}\gamma_{m-k}+\sigma_{e}^{2}\delta_{m,0}$$
where $\gamma_{m}$ is the autocovariance function of $X_{t}$, $\sigma_{e}$ is the standard deviation of the input noise process, and $\delta_{m,0}$ is the Kronecker delta function. This method involves analyzing the eigenvalues of the matrix in the Yule-Walker equations to determine the optimal order of the AR model.

By selecting the appropriate method for determining the optimal order and using Burg’s method for parameter estimation, we can effectively apply an AR model to LFP signals for biometric identification or other applications.

#### 2.3.2. Frequency Domain

As mentioned above, brain signals can be separated into different frequency bands, each of which is related to various brain activities. Converting LFP data into the frequency domain allows for the extraction and distinction of the dominant frequency components.

Power Spectral Density (PSD) is a useful measure that describes the signal strength distribution in the frequency domain. Fourier Transform (FT) is an effective method for transforming EEG signals from the time domain into the frequency domain. Based on the PSD obtained through squaring the absolute value of Fourier-transformed data in each segment, several LFP features can be calculated for further recognition purposes:**Mean/Variance of power spectrum**: The variance of spectral power is calculated by:
(4)$$\sigma^{2}=\frac{1}{N}\sum_{i=1}^{N}{\left(x_{i}-\bar{x}\right)}^{2}$$
where $x_{i}$ is a spectral power at each frequency bin, *N* is the number of frequency bins, and $\bar{x}$ is an average of all spectral powers. These features $\bar{x}$ and $\sigma^{2}$ measure the dispersion of the power spectrum (PS), which can help differentiate between different individuals.**Energy**: The energy of signals is computed with the Parseval’s spectral power ratio theorem:
(5)$$E\left(s\right)=\frac{1}{N}\sum_{n=1}^{N}s_{n}^{2}$$
where $s_{n}$ is the *n*-th sample of signal *s* and *N* is the total number of samples in the signal. This feature reflects the power intensity of the brain signals.**Concavity of spectral distribution**: The maximum of the power spectrum is detected and then its part is calculated and adopted as a criterion. Then the frequencies of which power spectral values are under the criterion are squared and then summed as:
(6)$$F_{u}=\sum_{j=1}^{N}{\left(f_{j}^{u}\right)}^{2}$$
where $f_{j}^{u}\left(j=1,2,\cdots ,N\right)$ is frequencies under the criterion. This feature captures the shape of the spectral distribution, which can provide information about the underlying brain activity [30].**Nondominant region of the power spectrum**: The non-dominate region of the signal is defined as follows. Firstly, the maximum power spectrum of the signal is detected and then a threshold is determined in proportion to the maximum power. Comparing a spectral power at each frequency bin with the threshold, if the spectral power is under the threshold, such a frequency is regarded as within the non-dominant region. All spectral powers within the non-dominant region are accumulated and then it becomes another spectral feature. The two spectral features are fused as a feature vector and the fusion is given by:
(7)$$\lambda =a_{1}\times \sigma^{2}+a_{2}\times l$$
where *l* is the total spectral power in the non-dominant region, and $a_{1}$:$a_{2}$ is the fusion ratio. This feature focuses on the less dominant frequency components, which can provide additional information for recognition tasks [31].

By utilizing these features based on PSD, we can obtain a comprehensive representation of the LFP signals, which can be useful for various applications, such as biometric identification and brain-computer interfaces.

#### 2.3.3. Time-Frequency Domain

Discrete Wavelet Transform (DWT) is a wavelet transform algorithm that provides both time and frequency information on the signals. Compared with DWT, Wavelet Packet Decomposition (WPD) is more robust as it decomposes both the detail and approximation coefficients, resulting in a more comprehensive representation of the signal. Specifically, WPD builds the complete wavelet packet tree by passing the signal through more filters, while in DWT, only the previous approximation coefficients are used to pass through quadrature mirror filters. Reference [32] used a 4-level WPD to separate EEG signals into 5 subbands (delta, theta, alpha, beta, and gamma) and extracted features such as mean and standard deviation values. By using WPD and Daubechies 4 wavelets, we can obtain a more detailed and robust representation of the LFP signals, which can be useful for various applications, such as biometric identification and brain-computer interfaces.

### 2.4. Classifiers

#### 2.4.1. Similarity-Based Algorithm

Similarity-based pattern recognition is indeed a classification approach used for authentication or identification of individuals based on selected similarity evaluation metrics [33]. K-Nearest Neighbors (KNN) is a commonly used algorithm for identification. KNN makes final decisions based on the majority rule, considering the closest or most similar points to the input data. By using similarity-based pattern recognition, we can effectively authenticate or identify individuals in various applications, such as biometric identification.

#### 2.4.2. Discriminant Analysis

Discriminant Analysis (DA), more specifically Linear Discriminant Analysis (LDA), is a dimensionality reduction and classification technique that aims to separate data from different classes by projecting them into a lower-dimensional space. The main goal of LDA is to maximize the inter-class distance while minimizing the intra-class distance. By using LDA, we can effectively separate and classify data in various applications, such as biometric identification, pattern recognition, and dimensionality reduction tasks.

#### 2.4.3. Support Vector Machine

An SVM (Support Vector Machine) is a powerful classification algorithm that uses a hyperplane to separate two classes of data by maximizing the margin, which is the distance between the nearest training points from different classes. SVMs have good generalization capabilities. Kernels in SVM are divided into linear kernel and nonlinear kernels. Linear kernel is computationally efficient, while nonlinear kernels are introduced to map data to another space to make them more separable and the classifier’s complexity is increased. One frequently used nonlinear kernel is the Radial Basis Function (RBF) kernel. By using SVM with linear or nonlinear kernels, we can effectively classify data in various applications, such as biometric identification and pattern recognition.

#### 2.4.4. Neural Network

Neural Networks (NN) are indeed one of the most important and popular machine learning techniques for mapping inputs to outputs. For instance, researchers utilized different neural networks to analyze medical signals and achieved good performance [34,35,36]. The classic structure for implementing an NN is the multilayer perceptron (MLP), which generally has three types of layers: input layer, hidden layer, and output layer. It uses the feedforward and back-propagation algorithm to train the data and calculate the weight matrix. Based on the weight matrix, the result can be predicted. Additionally, Deep Neural Network (DNN) is an extension of MLP with two or more hidden layers. DNN can capture more complex patterns and representations in the data. By using NNs, such as one-hidden layer NNs or Deep Neural Networks (DNNs), we can effectively classify and recognize patterns in various applications, including brain biometric recognition and other machine learning tasks. In this paper, we designed a simple neural network of three layers, two hidden layers with 50 and 30 neurons respectively, and an output layer of 10 neurons (corresponding to the 10 rats). For the training of the neural network, we selected the ReLU activation function and L2 regularization, and we used a learning rate of 0.001 and a total of 200 iterations with cross-entropy loss function. Additionally, we chose the Adam method for stochastic optimization.

## 3. Experiments and Results

### 3.1. Surgery

Due to the difficulty of collecting intracranial brain signals in normal people, we designed surgeries and experiments on rats. For these animal experiments, 10 adult male Sprague-Dawley rats (300–350 g) were used. Note that all surgical and experimental procedures in the Guide for The Care and Use of Laboratory Animals (China Ministry of Health) were strictly followed in this study, and our experiments were approved by the Animal Care Committee of Zhejiang University, China.

Rats were anesthetized with propofol (10 mg/mL, i.p., 1 mL/100 g initial dose) and mounted on a standard stereo-taxic apparatus (RWD Life Science, Shenzhen, China) for brain surgery. The body temperature was retained with a heating pad, with the heart rate (300–400 bpm) and pO2 (>90%) monitored during the surgery. The state of anesthesia was examinated by toe-pinch test at regular intervals. Additional dose of propofol (10 mg/mL, i.p., 0.6 mL) was injected if necessary. A 16-channel (2 × 8) handmade microelectrode array (35 µm nichrome) was implanted of which the anterior 2 × 4 electrodes were in rostral forelimb area (RFA) and posterior 2 × 4 lied in ipsilesional caudal forelimb area (CFA) with a depth of 1.2–1.5 mm, while the electrodes were attached to the skulls with tiny screws and dental cement.

Rats were trained to perform a running behavior task. A 80cm×9cm×12cm treadmill was utilized to encourage the rats to run. Here, the speed was set to 10 m/min. The rats were recovered for three to four weeks before training and routine experiments. The signal acquisition lasted for two weeks. For each experiment day, we collected five minutes of running data for each rat. The data were inspected visually to remove periods in which the rats were not running. All data were recorded using a commercial multi-channel neural signal acquisition system (Plexon TM, OmniPlex/128) with amplification of 1750.

### 3.2. Feature Optimization

In this paper, we utilized features of three domains for evaluation: time domain, frequency domain and time-frequency domain. The input signals are 2-s long filtered LFP samples for feature extraction as mentioned in Section 2.2.

Firstly, we employed AR features to represent the time domain features. As mentioned above, we attempted to minimize the error of the predictor equation through experimental results with different orders and minimize the Akaike Information Criterion (AIC) with Burg’s method to find the optimal order of the AR model that balances model complexity and prediction accuracy. In addition, we also compared the results based on the eigenvalues of the matrix $\tilde{R}$ in the Yule-Walker equations and we found that Burg’s method is relatively better in performance. Finally, we adopted an AR model of order 4 considering the balance of computation complexity and performance and we took the coefficients of the AR model as features, which are a 64-dim (4 × 16 channel) vector.

Secondly, for frequency domain features, we used fast fourier transform to obtain the frequency distribution of input signals. Then we tried four features of five bands (Delta, Theta, Alpha, Beta and Gamma) as described in Section 2.3.2 for selection of frequency domain features, that is, the mean and standard deviation values of power spectrum, energy values, concavity of spectral distribution and nondominant region of the power spectrum. After comparing the identification performance of these four features, we chose the mean and standard deviation values of power spectrum and energy features to represent the frequency domain features, which are a 160-dim (10 × 16 channel) vector and a 80-dim (5 × 16 channel) vector respectively.

Thirdly, DWT features and WPD features were selected to represent the time-frequency domain features. Specifically, we compared different wavelets to receive the best performance, such as Daubechies wavelets, Coiflets wavelets and Symlets wavelets. Additionally, we tried different number of iterations for DWT features and different number of decomposition levels for WPD features to obtain the best results. After comparison, for DWT features, we decomposed the signals using Daubechies 4 wavelets for 5 iterations and calculated the standard deviation values of the decomposed signals, which are a 96-dim (6 × 16 channel) vector. As for WPD features, we used the Daubechies 4 wavelets at level 3 with Shannon entropy and estimated the coefficients as features, which are a 192-dim (12 × 16 channel) vector.

Finally, we applied above 5 features of three domains to evaluate identification performance: T-AR features (time domain), F-PS features (frequency domain), F-Energy features (frequency domain), TF-DWT features (time-frequency domain) and TF-WPD features (time-frequency domain).

### 3.3. Intra-Day Identification

Firstly, we tried to evaluate the identification performance within day. Specifically, we adopted the 80% data of an experimental day for training and the rest 20% data of the same day for testing. Here we utilized the SVM classifier with linear kernel for computing identification accuracy. The results are represented in Table 1. For time domain features, T-AR features achieve 84% average identification accuracy of 14 days. For frequency domain features, F-PS and F-Energy features obtain 87% and 98% average identification accuracy of 14 days respectively. For time-frequency domain features, TF-DWT and TF-WPD features reach 96% and 97% average identification accuracy of 14 days separately. With these statistics, it is obvious that frequency domain and time-frequency domain features have higher performance than time domain features, which shows that features of intracranial brain signals related with frequency bands have higher reliability and more effective information than time domain features.

Additionally, F-Energy features have better performance than time-frequency domain features (TF-DWT and TF-WPD). Specifically, the identification accuracy of F-Energy is above 95% for all 14 days, compared with 10 days of TF-DWT and 11 days of TF-WPD, which represents the better reliability of F-Energy features. This result might indicate that energy of different frequency bands are more effective and stable for practical applications.

### 3.4. Inter-Day Identification

Furthermore, we designed the inter-day identification experiments to testify the capability of the training model using previous days data to predict the signals of new days. Here we adopted the first day for training and the last 13 days for testing, the results of 5 features are shown in Table 2. It is obvious that the identification accuracy is slowly descending along with the days for all 5 features. We take F-Energy as an example, the accuracy of day-2 and day-3 can achieve 80% and 82%, while the accuracy of day-12 and day-13 drops to 51% and 56%. The difference between the test accuracy of previous days and following days may due to the electrode drifts. In fact, the implantable electrodes in rats are not in constant positions owing to the behaviors of rats. With a tiny change of the position, the collected signals can be quite different, which causes the identification errors of the training model. However, if we utilize the training data and testing data in the same day, this question can be handled. With the days grow, the position changes of the electrodes are larger and the training model is more inaccurate. Among 5 features, F-Energy feature has the highest average identification accuracy of 67%, which represents that F-Energy feature is more reliable in practical use.

Moreover, we think that the reasons for low inter-day identification accuracy could be the simplicity of training models. Therefore, we attempted to adopt more training days and evaluate the identification performance of lasting days. As shown in Figure 2, we utilized training days from 1 to 13 and computed the average identification accuracy of lasting days. The results show that as time increases, the accuracy is raising for all 5 features. Significantly, if we take 13 days for training, the identification accuracy of F-Energy feature can obtain 93% (91% for both TF-DWT and TF-WPD) as shown in Table 3, which is relatively high. These results confirm our assumptions that with more training data, the inter-day identification performance of intracranial brain signals can reach higher.

### 3.5. Performance of Different Classifiers

Except for the SVM classifier with linear kernel, we also designed experiments with other machine learning algorithms, such as KNN, LDA, SVM with RBF kernel and Neural Network. For the KNN method, a k-value of three was selected by comparison with the performance of different k-values to yield the best performance. While for the LDA and SVM-RBF methods, we adopted the standard implementation. Additionally, we designed a neural network of three layers, two hidden layers with 50 and 30 neurons respectively, and an output layer of 10 neurons. Specifically, we selected the ReLU activation function and L2 regularization, and we used a learning rate of 0.001 and a total of 200 iterations with cross-entropy loss function for training. Additionally, we chose the Adam method for stochastic optimization. Here we realized the previous four classifiers with the Matlab toolbox (Matlab 2021b version). As for the implementation of Neural Network, we utilized pytorch models.

Here we chose T-AR, F-Energy and TF-DWT features as input to compare the intra-day identification performance of three domain features with 5 different classifiers. For T-AR features, LDA, SVM-Linear and Neural Network have similar performance with the average 85% identification accuracy, while the average accuracy of KNN and SVM-RBF is 82% and 59% respectively as shown in Figure 3. Similarly, as shown in Figure 4, LDA, SVM-Linear and Neural Network have the best performance with the average 98% identification accuracy of F-Energy features, while the average accuracy of KNN and SVM-RBF is 91% and 25% respectively. As for TF-DWT features, as shown in Figure 5, KNN algorithm and Neural Network have the best performance with the average 98% identification accuracy, while the average accuracy of SVM-Linear, LDA and SVM-RBF is 96%, 89% and 62% respectively. These results show that neural network is optimal for identification of brain biometrics, and SVM-Linear classifier is also fine. For the reason of low accuracy of SVM-RBF may be that the input features might have linear correlation or the training data is not enough for training SVM-RBF. Here we recommend researchers to use neural networks for higher performance with more computation load or to adopt SVM-Linear classifier for time efficiency.

## 4. Discussion

In this paper, we collected the local field potential signals for identification to analyze the features of intracranial brain signals. Specifically, we adopted three domain features for evaluation: time domain features, frequency domain features and time-frequency domain features. Similarly, researches on EEG signals also use these types of features for person identification. For instance, Autoregressive (AR) model is a widely used time-domain feature in EEG biometrics and many researchers adopted AR features for identification [27,28,29]. Compared with their single approach to select the optimal model order *p*, we utilized three optimization algorithms to determine the order *p* to obtain the best performance. Additionally, researchers usually chose Power Spectral Density (PSD) to describe the signal strength distribution in the frequency domain for EEG biometrics [30,31]. By comparison, due to the higher resolution and signal-to-noise ratio (SNR) of LFP signals used in this paper, we could obtain more accurate frequency distribution than EEG signals and the identification performance (98% intra-day accuracy and 93% inter-day accuracy of energy features) is relatively better. With the same reason, for time-frequency domain features, we could reach better performance (97% intra-day accuracy and 91% inter-day accuracy of WPD features) than EEG signals.

Additionally, we also compared the performance of 5 features of different domains using 5 different classifiers. The results show that frequency domain features and time-frequency domain features are better than time domain feature in intra-day and inter-day performance. The reason for this may be that the time domain features are linear features which are directly related to the raw brain waves. It is obvious that brain waves are changing all the time, leading to the AR model is not always adequately accurate for identification. Differently, frequency related features are based on particular bands, which are relatively more stable. In addition, energy features have best identification performance among 5 features. From this aspect, we think that energy of different frequency bands might reflect intrinsic characteristics of brain waves, which contains the most effective information in brain system. Furthermore, we find that time-frequency domain features also perform well in both intra-day and inter-day experiments. Moreover, we testified 5 different classifiers and found that Neural Network and SVM-Linear have higher performance. For reaching higher evaluation metrics, we recommend Neural Network, which is complex enough for different data; while for the time efficiency, we recommend SVM-Linear for achieving similar performance of Neural Network but is faster for training and testing. After all, we think that energy feature and time-frequency domain features are excellent for biometric identification, and we put forward Neural Network and SVM-Linear for training model of these intracranial brain features.

Furthermore, our research emphasis is on intracranial brain signals due to these signals have better resolution and signal-to-noise ratio (SNR) than invasive signals (such as EEG), so we think that how to collect intracranial brain signals in invasive or harmless ways of people is our future target. Nowadays, flexible electrodes have been experimented in invasive brain signal collection, such as the threads of Neuralink, which can obtain the accurate intracranial brain signals with minor damage on the brain surface. In this case, we think that it would be convenient and harmless to collect intracranial brain signals with the development of the flexible electrodes or other collection materials. In recent days, the Neuralink company has received the FDA’s approval to launch their first-in-human clinical study. Hence we think this might be ethically correct if the technology is used in right ways.

## 5. Conclusions

In this paper, 5 features based on intracranial brain signals in three domains are computed for identification, and their performance is compared with different machine learning algorithms. The results show that frequency features and time-frequency features are excellent both for intra-day and inter-day identification. Additionally, energy features obtain best identification performance among 5 features with 98% intra-day and 93% inter-day identification accuracy. Moreover, we testified 5 different classifiers and found that Neural Network and SVM-Linear have higher and more stable performance. To the best of our knowledge, this is the first study to investigate the features of intracortical brain signals for identification and we hope this research can serve as a guidance for future intracranial brain research and the development of more reliable and stable brain-based biometrics. In future studies, we intend to optimize the methods to improve the inter-day identification performance with reducing the noise of electrode drift. Furthermore, we plan to collect the intracranial brain signals of human beings if possible and evaluate the performance of three domain features.

## Figures and Tables

**Figure 1 bioengineering-10-00801-f001:**
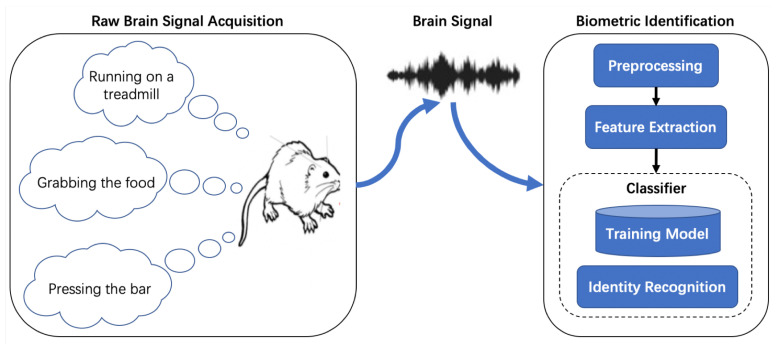
Structure of brain biometric identification system.

**Figure 2 bioengineering-10-00801-f002:**
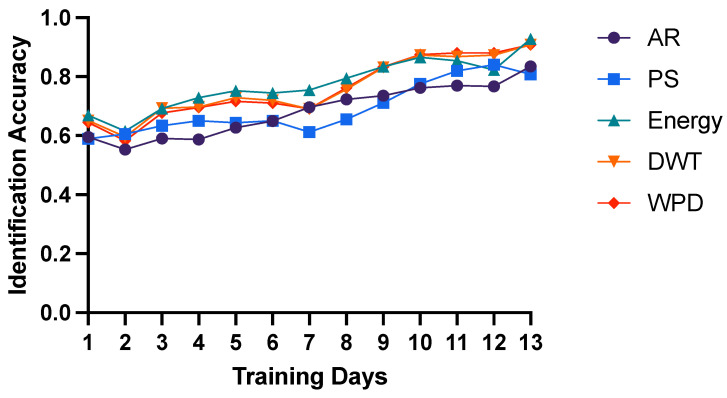
Inter-day identification performance with different number of training days from 1 to 13 of 5 features using SVM classifier of linear kernel.

**Figure 3 bioengineering-10-00801-f003:**
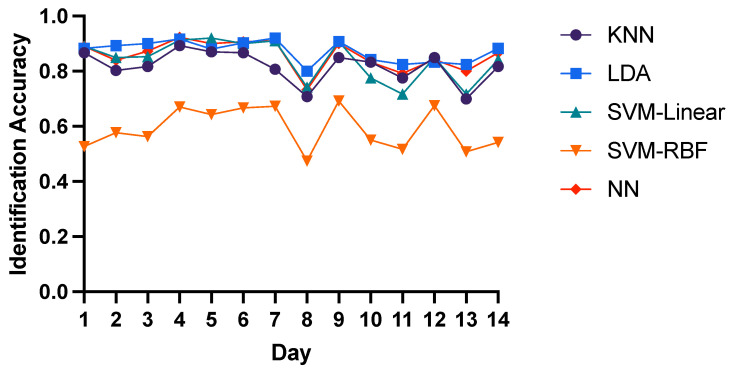
Intra-day identification accuracy of T-AR features.

**Figure 4 bioengineering-10-00801-f004:**
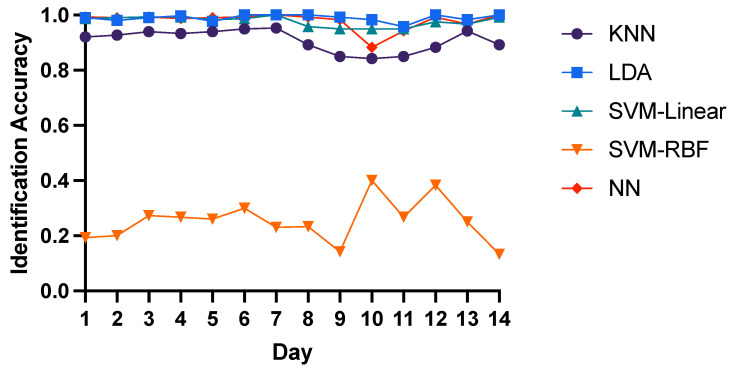
Intra-day identification accuracy of F-Energy features.

**Figure 5 bioengineering-10-00801-f005:**
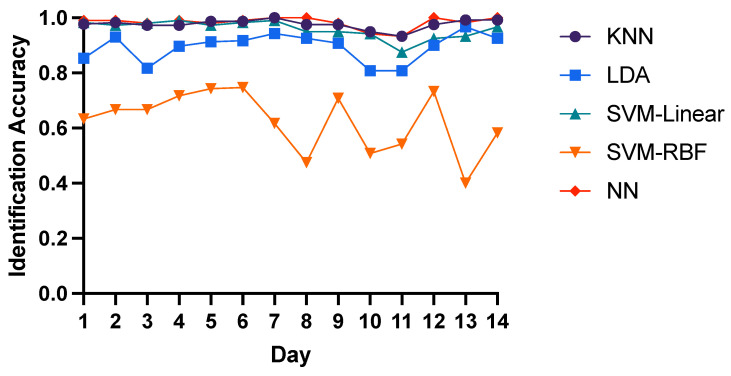
Intra-day identification accuracy of TF-DWT features.

**Table 1 bioengineering-10-00801-t001:** Intra-day Identification Performance of 5 Features using SVM (Linear).

Features	1	2	3	4	5	6	7	8	9	10	11	12	13	14	Avg
**T-AR**	0.89	0.85	0.85	0.91	0.92	0.90	0.91	0.74	0.91	0.78	0.72	0.85	0.72	0.84	**0.84**
**F-PS**	0.91	0.95	0.92	0.93	0.94	0.90	0.97	0.87	0.80	0.76	0.69	0.78	0.88	0.90	**0.87**
**F-Energy**	0.99	0.99	0.99	0.99	0.98	0.99	1.00	0.96	0.95	0.95	0.95	0.98	0.97	0.99	**0.98**
**TF-DWT**	0.98	0.97	0.98	0.99	0.97	0.98	0.99	0.95	0.95	0.94	0.88	0.93	0.93	0.97	**0.96**
**TF-WPD**	0.98	0.98	0.98	0.99	0.97	0.99	0.99	0.98	0.96	0.94	0.88	0.93	0.95	1.00	**0.97**

**Table 2 bioengineering-10-00801-t002:** Inter-day Identification Performance of 5 Features using 1 Training Day.

Features	1	2	3	4	5	6	7	8	9	10	11	12	13	Avg
**T-AR**	0.61	0.74	0.74	0.70	0.62	0.59	0.50	0.60	0.54	0.52	0.55	0.48	0.57	**0.60**
**F-PS**	0.61	0.70	0.73	0.70	0.67	0.65	0.58	0.50	0.43	0.45	0.50	0.57	0.57	**0.59**
**F-Energy**	0.68	0.80	0.82	0.78	0.72	0.73	0.62	0.66	0.57	0.61	0.63	0.51	0.56	**0.67**
**TF-DWT**	0.70	0.79	0.81	0.79	0.75	0.68	0.59	0.61	0.54	0.58	0.62	0.52	0.49	**0.65**
**TF-WPD**	0.68	0.80	0.81	0.78	0.70	0.68	0.56	0.59	0.53	0.55	0.61	0.53	0.56	**0.64**

**Table 3 bioengineering-10-00801-t003:** Inter-day Identification Performance of 5 Features using 13 Training Days.

Features	Test Day
**T-AR**	**0.84**
**F-PS**	**0.81**
**F-Energy**	**0.93**
**TF-DWT**	**0.91**
**TF-WPD**	**0.91**

## Data Availability

The datasets are available on reasonable requests from corresponding author.

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
