# Peer review of "Optimal Feature Analysis for Identification Based on Intracranial Brain Signals with Machine Learning Algorithms"

_bioengineering, 2023, doi:10.3390/bioengineering10070801_

Round 1

Reviewer 1 Report

Authors proposed the features of intracortical brain signals for identification and their performance is compared with different machine learning algorithms. Here are some of my comments.

1.      Optimal feature analysis  as stated in the title. It describes the computation of several features based on local field potential for identification, but it does not discuss the optimality or selection process of these features.

2.      How are the “Energy Features” defined? how reliable are they for intra-day and inter-day identification?

3.      How are the features extracted from LFP signals in the time-frequency domain? What techniques or algorithms are typically used to capture the joint time-frequency characteristics of the signals?

4.      Define "non-dominant region" in Eq. 7.

5.      Details about the architecture and configuration of the neural network with three layers are to be described such as what activation functions were used, were there any regularization techniques employed etc.

Language usage is good. 

Reviewer 2 Report

The work "Optimal Feature Analysis for Identification Based on Intracranial Brain Signals with Machine Learning Algorithms" by Ming Li et al. presents a novel and provocative idea of using intercranial local field potential to extract specific signatures which could serve as biommetric markers. Although the novelty there are some issues that requiere to be addresed before the acceptance of this paper, for instance:   1) The research has done using rats. 2) Intracortical brain signal, need electrodes to be implanted directly inside the brain, Could you address in the introduction, discussion and conclusions, how are your planning to do this in humans in a large scale? is this ethically correct? 3) The authors claim that this brain activity measuring technique is better than Surface EEG tecniques. However, authors found a similar problems encounter when using surface EEG tecniques, noise contamination which results in low SNRs. The authors needed to do extra preprocessing to clean up all the noise first. This problem also emerges in section 3.4 Inter-day identification, the difference between the test accuracy of previous  days and following days  may due to the electrode drifts. In fact, the electrodes in the rats are not in constant positions owing to the behaviors of the rats. With a tiny change of the position, the collected signals can be quite different, which causes the identification errors of the training model. This a common problem encounter while using surface EEG electrodes, what is the advantange of using intracranial electrodes then? The authors need to to do abetter job in the introduction to make their point of using intracranial electrodes stronger. As it is, readers won't buy the idea. 4) Figure 1 is misleading, the experimentation was done in rats not humans, please correct it with the real experimental setup.

Round 2

Reviewer 2 Report

After reading the new version the author have properly addressed all points I raised in my previous interaction. I think the article can be published in its present form.